# Activation of Adenosine Triphosphate-Gated Purinergic 2 Receptor Channels by Transient Receptor Potential Vanilloid Subtype 4 in Cough Hypersensitivity

**DOI:** 10.3390/biom15020285

**Published:** 2025-02-14

**Authors:** Wanzhen Li, Shengyuan Wang, Tongyangzi Zhang, Yiqing Zhu, Li Yu, Xianghuai Xu

**Affiliations:** Department of Pulmonary and Critical Care Medicine, Tongji Hospital, School of Medicine, Tongji University, Shanghai 200065, China

**Keywords:** cough hypersensitivity, transient receptor potential vanilloid 4, adenosine triphosphate, purinergic receptors, signaling pathway

## Abstract

Background: Transient receptor potential vanilloid subtype 4 (TRPV4) is a Ca^2+^-permeable non-selective cation channel that is involved in the development of cough hypersensitivity. Purinergic 2 receptors (P2X) belong to a class of adenosine triphosphate (ATP)-gated non-selective cation channels that also play an important role in cough hypersensitivity. Nevertheless, little is known about the interaction between them for cough hypersensitivity. The present study was designed to clarify the roles of TRPV4 and ATP-P2X receptors in cough hypersensitivity, and to explore the possible involvement of ATP-P2X receptors in the development of cough hypersensitivity mediated by TRPV4. Design and Method: This study aims to establish a guinea pig model of citric acid-induced enhanced cough to confirm the effects of the TRPV4-mediated purinergic signaling pathway on cough sensitivity by testing the number of coughs, the release of ATP, and the expressions of P2X and TRPV4 receptors in the tracheal carina and vagal ganglion; recording the activity of cellular currents with the whole-cell patch clamp technique; and detecting changes in intracellular calcium flow in the vagus nerve cells. Results: The number of coughs in the TRPV4 agonist GSK1016790A-treated control group was elevated compared with that in the control group, whereas the number of coughs in the TRPV4 antagonist HC067047-treated model group was significantly reduced compared with that in the chronic cough group. When the individuals in the chronic cough group were treated with A317491, PSB12062, and A804598 (P2X3,4,7 antagonists), the number of coughs was significantly decreased. This suggests that TRPV4 and P2X3, P2X4, and P2X7 receptors have an effect on cough hyper-responsiveness in guinea pigs with chronic cough. Enzyme-linked immunosorbent assay results suggested that TRPV4 antagonist and P2X3,4,7 antagonist could differentially reduce the levels of inflammatory factor SP and CGRP in alveolar lavage fluid, and TRPV4 antagonist could reduce the ATP content in the alveolar lavage fluid of guinea pigs in the model. Western blot and immunohistochemistry results showed that, in the tracheal carina and vagal ganglion, the TRPV4 and P2X3,4,7 expression was elevated in the chronic cough group compared with the control group, and could be significantly inhibited by TRPV4 antagonist. Vagus ganglion neurons were isolated, cultured, identified, and subjected to whole-cell membrane clamp assay. When ATP was given extracellularly, a significant inward current was recorded in the examined cells of individuals in the chronic cough and control groups, and the inward current induced by ATP was higher in the chronic cough group relative to the control group. This inward current (I_ATP_) was differentially blocked by P2X3, P2X4, and P2X7 antagonists. Further studies revealed that TRPV4 agonists potentiated ATP-activated currents, and the potentiated currents could still be inhibited by P2X3, P2X4, and P2X7 receptor antagonists, whereas TRPV4 inhibitors partially blocked ATP-activated currents. It is suggested that TRPV4 affects P2X3, P2X4, and P2X7 receptor-mediated ATP-activated currents. Calcium imaging also showed that TRPV4 agonists induced different degrees of calcium inward currents in the vagal neurons of the chronic cough and the control group, and the calcium inward currents were more significant in the model group. Conclusions: The TRPV4-mediated purinergic signaling pathway was identified to be involved in the development of cough hypersensitivity in guinea pigs with chronic cough; i.e., TRPV4 can lead to the release of airway epithelial ATP, which can stimulate P2X receptors on the cough receptor, and further activate the sensory afferent nerves in the peripheral airway, leading to increased cough sensitivity.

## 1. Background

Chronic cough (CC) is defined as a cough which lasts for over 8 weeks without abnormalities on chest imaging [1]. Among CC patients, 10–42% of them have unknown causes or lack effective treatments [2,3], and their persistent and refractory cough symptoms are categorized as refractory chronic cough (RCC), which has gradually attracted the attention of clinicians and become a hot issue in the diagnosis and treatment of CC.

It has been found that most patients with RCC have increased cough sensitivity, known as cough hypersensitivity syndrome (CHS) [4]. This syndrome is characterized by coughing being triggered by low-level stimuli, and is associated with peripheral and central sensitization [5]. Peripheral sensitization of coughing refers to the excitability of the peripheral terminals of vagal sensory fibers, leading to increased sensory input and subsequently triggering coughing [6]. There are two types of peripheral sensory innervation: myelinated Aδ fibers and unmyelinated C fibers, which are sensitive to mechanical and chemical stimuli, respectively, generating cough signals. These receptors can also interact with inflammatory cytokines, such as macrophage inflammatory protein-2 and interleukin-1b, through airway damage, exacerbating coughing and making it difficult to control [7].

The transient receptor potential (TRP) family is a class of non-selective cation channel proteins. Currently, many cough-related studies focus on the transient receptor potential vanilloid-1 (TRPV1) and transient receptor potential ankyrin-1 (TRPA1) expressed in C-fibers, which are considered to be closely related to CHS [8,9]. However, due to poor efficacy, side effects, and the problem of solvent-induced coughing, the effectiveness of TRPV1 and TRPA1 antagonists in inhibiting coughing requires further evidence [10,11,12,13]. Transient receptor potential vanilloid-4 (TRPV4) is another important member of the TRP subfamily V, expressed in peripheral ganglia, lung fibroblasts, alveolar macrophages, and smooth muscle cells in various organs [14,15,16,17]. Recent studies have shown that the TRPV4 channel is mechano-sensitive and osmotically sensitive, and its activation can lead to an increase in the concentration of intracellular calcium (Ca^2+^) [16], and, subsequently, a series of physiological and pathological processes, such as neurotransmitter release and excitation conduction. Studies have demonstrated that the small-molecule TRPV4 agonist GSK1016790A can induce coughing in healthy guinea pigs, which can be inhibited by the selective TRPV4 antagonist HC067047. The mechanism may involve an increased concentration of intracellular Ca^2+^ in the vagal ganglion and the depolarization of vagus nerve neurons [18]. In addition, activating TRPV4 receptor can also cause airway smooth muscle contraction in humans and animals, possibly through the activation of Aδ afferent nerve fibers [19]. In a pain-related research, TRPV4 has also been shown to promote neurogenic inflammation and pain hypersensitivity through Ca^2+^ and CGRP [20,21,22]. Therefore, TRPV4 may be involved in the formation of cough hypersensitivity, and antagonizing TRPV4 may represent a potential new therapeutic target for cough. However, there are currently few clinical studies on the use of TRPV4 antagonists for cough treatment, which necessitates further exploration.

Extracellular adenosine triphosphate (ATP) interacts with purinergic 2 receptors (particularly, P2X) and plays a regulatory role in the chronic sensitization of peripheral nerves. It is currently believed to be associated with neuropathic pain, itch, CC, and other diseases. ATP-mediated P2X receptors, especially P2X3 receptor, have been a research hotspot in recent years with regard to RCC, but the application of P2X3 receptor is limited, due to insufficient efficacy and taste disturbances [23,24]. Other P2X receptor subtypes, such as P2X4 and P2X7 antagonists, have shown significant efficacy in rat models of neuropathic pain [25,26]. However, fewer studies have explored their efficacy against cough hypersensitivity, and there are some challenges in terms of their efficacy, receptor selectivity, and pharmacokinetics. Herein, indirect inhibition of P2X receptors by targeting upstream may be an effective approach. Studies have shown that the activation of TRPV4 channels in urothelial cells regulates ATP release, modulating the mechanical sensitivity of bladder afferent nerves [27]. The interaction between TRPV4 and AQP2 influences ATP release in cortical collecting duct cells [28]. In rat pulmonary endothelial cells, TRPV4 channel activation can release ATP via pannexin (PANX1) hemichannels, which acts on P2Y receptors, causing Ca^2+^ oscillations [29]. In a cough model, Bonvini et al. demonstrated that the depolarization of isolated vagal nerves induced by a TRPV4 agonist can be inhibited by a P2X3 receptor antagonist, which specifically targets ATP [18]. We thus hypothesized that the mechanism of TRPV4-mediated cough hypersensitivity may be associated with ATP-P2X.

In this research, which investigates the role of the TRPV4-mediated purinergic signaling pathway in cough hypersensitivity, we study the P2X change in a citric acid-induced enhanced cough model in response different pharmaceutical interventions, and explore the interaction between TRPV4 and ATP-P2X.

## 2. Materials and Methods

### 2.1. Chemicals

Citric acid (optimal concentration = 0.4 M, Sigma-Aldrich, St. Louis, MO, USA) and pentobarbitone (recommended concentration = 20 mg/mL) were dissolved in saline. GSK1016790A (TRPV4 agonist, recommended concentration = 30 μg/mL, MCE, Shanghai, China), HC067047 (TRPV4 antagonist, optimal concentration = 30 mg/kg, 3 mg/mL, MCE), NF449 (P2X1 antagonist, optimal concentration = 3.0 μM, MCE, Shanghai, China), PSB10211 (P2X2 antagonist, optimal concentration = 10 μM, MCE, Shanghai, China), A317491 (P2X3 antagonist, optimal concentration = 5.0 μM, MCE, Shanghai, China), PSB12062 (P2X4 antagonist, optimal concentration = 10.0 μM, MCE, Shanghai, China), A804598 (P2X7 antagonist, optimal concentration = 8.0 μM, MCE, Shanghai, China), BzATP (P2X antagonist, optimal concentration = 6 mg/kg, MCE, Shanghai, China), and ATP (optimal concentration = 10^−4^ mM, MCE, Shanghai, China) were dissolved in DMSO. On the day of the experiment, the final concentration dilutions of all chemicals were prepared in normal saline and dimethyl sulfoxide. Both the doses and administration schedules were selected based on previous studies and pilot experiments, and methods of minimizing side effects were also considered.

### 2.2. Animals and Citric Acid-Induced Enhanced Cough Model

General-grade Hartley male guinea pigs with a weight of 300–350 g were purchased from Shanghai Jieshijie Laboratory Animal Centre. The Laboratory Animal Centre of Shanghai Tongji Hospital is responsible for rearing. Guinea pigs were required to undergo one week of acclimatization before undergoing nebulized inhalation experiments. The TRPV4 knockout mice were purchased from Saiye Biotechnology Co., Ltd. in Suzhou, China. These procedures were approved by the Laboratory Animal Ethics Committee of Shanghai Tongji Hospital (2023-DW-(048)), and were conducted in strict compliance with the principles and standards of international guidelines for the protection and use of laboratory animals [30].

The establishment of the guinea pig citric acid-induced enhanced cough model (chronic cough model or CC model for short) was based on a previous study [31] conducted by our research group, as follows: awake guinea pigs were placed in an EMKA non-invasive lung function test system Device (EMKA, Paris, France) with a whole-body plethysmography system. Using a nebulizer (with aerosol particle diameters of 0.5–2.0 μm and an atomization flow rate of 0.2 mL/min, EMKA, Paris, France), the guinea pigs inhaled a 0.4 M citric acid solution for 3 min to induce coughing, twice daily (firstly between 8:00 and 9:00 AM, and secondly between 8:00 and 9:00 PM) for a total of 15 days. During the process, the number of coughs and morphological and inflammatory changes in tracheal mucosa were detected to validate the effectiveness and reliability of the CC model.

The methodology of the GSK1016790A-induced cough in conscious guinea pigs referred to Bonvini et al.’s research [18]. A concentration response was initially established for GSK1016790A (30 μg/mL) or an appropriate vehicle (1% ethanol and 1% Tween 80 in saline). The stimulus was aerosolized for 5 min.

To assess cough reactivity, we used the Cough module of the EMKA system. During data processing, the system calculated two key metrics: the area under the airflow curve (V2) and the half peak conversion time of airflow (DHPC). These raw data were accurately converted using a simulation equation and analyzed alongside acoustic waveform changes, enabling the system to automatically identify and mark cough waveforms. The method was conducted as described by Canning et al. [32]. In preliminary experiments with healthy awake guinea pigs, capsaicin inhalation was used to determine concentration selection, with the number of coughs occurring within 3 min calculated (from the start of inhalation for 2 min to 1 min after cessation). Trained observers (2 in the room) were blind to the groups and counted the number of coughs, and a mean of the 2 observers’ counts was taken as the final count.

### 2.3. Enzyme-Linked Immunosorbent Assay

The substance P (SP) and calcitonin gene-related peptide (CGRP) levels in the bronchoalveolar lavage fluid (BALF) were determined in duplicate using DuoSet Enzyme-Linked Immunosorbent Assay (ELISA) kits with an assay range of 39.0–2500 pg/mL (R&D Systems, Minneapolis, MN, USA), according to the manufacturer’s instructions. The intra-assay and inter-assay variability of the measurements were 5% and 10%, respectively, across the range of concentrations measured. Every sample (50 µL) occupied three wells of the 96-well plate, and the data obtained represented an average of three measurements. Chemical methods were used to detect the ATP level in the BALF. Chemiluminescence values were measured with luminometer instruments.

### 2.4. Western Blotting

Western blotting (WB) was performed as described in previous studies. The sample of total protein was separated by 10% SDS-PAGE and then transferred to polyvinylidene fluoride membrane. The membrane was incubated in 5% milk for 2 h at room temperature. Next, the membrane was incubated with primary antibodies at 4 °C overnight, and afterwards with horseradish peroxidase (HRP)-conjugated secondary antibodies for 1 h at room temperature. The primary antibodies used were as follows: rabbit anti-TRPV4 antibody (1:5000, Abcam, Cambridge, UK), mouse anti-P2X3 antibody (1:500, Santa Cruz, Shanghai, China), rabbit anti-P2X4/7 antibody (1:1000, Proteintech, Wuhan, China), and rabbit anti-GAPDH antibody (1:5000, Proteintech, Wuhan, China). The secondary antibodies used were as follows: HRP-conjugated anti-rabbit (1:10,000, APExBIO, Shanghai, China) and anti-mouse (1:10,000, APExBIO, Shanghai, China). The protein signal was visualized using chemiluminescence (Immobilon Western HRP Substrate; Merck Millipore, Darmstadt, Germany), in accordance with the manufacturer’s instructions. The protein bands were detected with a MiniChemi imaging system (BioImaging Systems, United States), and quantification of the intensities of the bands was performed with the Image J software (1.53c), and normalized to ACTIN. The controlled experiments of P2X3 and P2X4 for the Western blots and related literature on P2X7 are shown in the Appendix A.

### 2.5. Immunohistochemical Analysis

Immunohistochemistry (IHC) was conducted to analyze the expression and distribution of the P2X3/4/7 protein in the tracheal carina and vagus nerve ganglion tissue. Firstly, 4% paraformaldehyde-fixed tracheal carina and vagus ganglion tissues were dehydrated, embedded, sectioned, and deparaffinized overnight. Endogenous peroxidases were inhibited with 0.5% hydrogen peroxide in methanol for 10 min, followed by overnight incubation at 4 °C with a rabbit polyclonal IgG antibody against P2X3/4/7 (1:500; Proteintech, Wuhan, China). The washed sections were then incubated for 1 h at room temperature with goat anti-rabbit IgG conjugated with horseradish peroxidase (HRP; 1:5000, Sigma-Aldrich, St. Louis, MO, United States). Visualization was performed by using 3, 3′-diaminobenzidine tetrahydrochloride (DAB) for 3 min and viewing the samples under a light microscope at 400× magnification. For each of the five animals in every group, a section was randomly chosen, and five fields were randomly selected from each section. The Image Pro Plus 6.0 system (Media Cybernetics, MD, Rockville, MD, USA) was used to detect the integral optical density (IOD) of positively stained sections (brown staining) and identify P2X3/4/7 in each field within an area of 25 µm^2^, including the four corners and central area. The total area of the region of interest (the epithelial layer) and the IOD were measured objectively. The average quantitative value of five fields was used for statistical analysis.

### 2.6. Cell Dissociation and Calcium Imaging

Guinea pigs were killed by means of injection of pentobarbitone (50 mg/kg administered intraperitoneally). Nodose ganglia were dissected free of adhering connective tissue, and neurons were isolated by means of enzymatic digestion, as described previously [33].

A total of 2 mL of Lockes buffer, 2 μL of Ca^2+^ indicator Fluo-4 (1 mM), and 4 μL of 5% Pluronic F-127 were added to a Petri dish with neurons attached, and incubated at 37 °C for 30 min to allow the cells to be fully loaded with the Ca^2+^ indicator. The slides containing the cells were transferred to a dedicated recording slot on the bench of the inverted microscope, and extracellular fluid was continuously perfused into them at a rate of 8 mL/min through a specific perfusion system. A suitable field of view was found under ordinary light and switched to a fluorescent light source. The fluorescence intensity of the cell Fluo-4 was detected by switching the filter module. The fluorescence was captured with a Leica DF350 camera, and the software recorded and analyzed the image signals in real time.

### 2.7. Cell Membrane Clamp Assay

Based on neuronal cell culture observations, we selected neurons after 48–72 h of culture for membrane clamp experiments. The extracellular perfusate composition (mM) was as follows: NaCl 150, KCl 5, CaCl_2_ 2.5, MgCl_2_ 1, HEPES 10 (Sigma-Aldrich, St. Louis, MO, USA), D-glucose 10, and pH adjusted to 7.3–7.4 with NaOH. The composition of intracellular fluid (mM) was as follows: KCl 140, MgCl_2_ 2, HEPES 10, EGTA 11, ATP 5, and pH adjusted to 7.2 with KOH. Whole-cell recordings were sampled at 10 kHz and filtered at 5 kHz for analysis (Axon 200B amplifier with pCLAMP software (11.4), Axon Instruments, Foster City, CA, USA). The electrode impedance was 3~5 MΩ. By using a certain amount of positive pressure, we avoided the electrode being contaminated with impurities. When the electrode was close to the cell, at a distance of 100~200 μm, the formation of high resistance between the electrode and the cell was sealed and stabilized with the use of the ZAP program (11.4). The suction broke through the cell, resulting in a whole-cell state. After the stabilization of the cell, recording started. The concentrations of ATP, A317491, PSB12062, A804598, GSK1016790A, and HC067047 were 10^−4^ mM, 10 μM, 30 μM, 8 μM, 100 μM, and 1 μM, respectively.

### 2.8. Statistical Analysis

GraphPad Prism 8.0 was used for plotting, Adobe Photoshop (20.x) for image assembly, and SPSS 23.0 for data analysis. When the homogeneity of variance assumption was met, a *t*-test was used to compare results between the two groups. If the homogeneity of variance assumption was not satisfied, the Mann–Whitney U test was applied. For comparisons among multiple groups, one-way analysis of variance (ANOVA) was conducted. If the ANOVA results indicated significant differences, the S-N-K post hoc test was employed for multiple comparisons. Data were expressed as the mean ± standard deviation, or the median (interquartile range), and a *p*-value < 0.05 was considered statistically significant.

## 3. Results

### 3.1. The Effects of TRPV4 and P2X Receptors on Cough Reactivity in Guinea Pigs

Repeated inhalation of 0.4 M citric acid led to an increase in the number of coughs in guinea pigs, peaking on day 15. Further inhalation resulted in a decrease in cough frequency, suggesting the presence of an adaptation mechanism to cough hypersensitivity. Additionally, on days 15 and 20 of citric acid inhalation, guinea pigs showed a significant increase in the proportion of neutrophils in BALF, as well as elevated levels of SP and CGRP, accompanied by injury of the tracheal mucosa (Appendix A). These findings confirm the successful establishment of a CC model in guinea pigs.

The cough frequency in the CC group was significantly higher than that in the control group (16 [12.5–17] coughs vs. 2 [0–2.5] coughs, *p* < 0.001). Nebulized inhalation of GSK1016790A (TRPV4 agonist) significantly increased the cough frequency in healthy control guinea pigs (15 [13–20] coughs vs. 2 [0–2.5] coughs, *p* < 0.001). Conversely, treatment with HC067047 (TRPV4 antagonist) significantly reduced the cough frequency in guinea pigs with CC (4 [3–7] coughs vs. 16 [12.5–17] coughs, *p* < 0.001) (Figure 1A).

When studying P2X receptor antagonists, it was found that nebulized inhalation of A317491 (P2X3 antagonist), PSB12062 (P2X4 antagonist), and A804598 (P2X7 antagonist) reduced the cough frequency in the CC group (2 [0.5–7] coughs vs. 15 [13–21] coughs, *p* = 0.001; 2 [0.5–7.5] coughs vs. 15 [13–21] coughs, *p* = 0.009; and 6 [3–9] coughs vs. 15 [13–21] coughs, *p* = 0.012, respectively) (Figure 1B). However, treatment with NF449 (P2X1 antagonist) and PSB10211 (P2X2 antagonist) showed no significant difference compared to the CC group.

We administered a combination of the TRPV4 agonist and P2X3, P2X4, or P2X7 antagonists to healthy control guinea pigs. The results indicated that the cough frequency significantly decreased in the GSK1016790A+A317491 group, GSK1016790A+PSB12062 group, and GSK1016790A+A804598 group, compared to the GSK1016790A-only group (3 [0–5.5] coughs vs. 15 [7.5–19.5] coughs, *p* = 0.010; 5 [2.5–8.0] coughs vs. 15 [7.5–19.5] coughs, *p* = 0.032; 6 [4–10] coughs vs. 15 [7.5–19.5] coughs, *p* = 0.036) (Figure 1C).

### 3.2. The Relationship Between ATP and Cough Hypersensitivity

During the establishment of the CC model, it was observed that ATP levels increased over time. On day 15, the ATP concentration in the airways reached its highest level (1.52 ± 0.27 vs. 0.15 ± 0.09, *p* < 0.001) (Figure 1D). On day 20, ATP levels plateaued, and the changes in ATP were consistent with the increase in cough reactivity. When control guinea pigs and CC guinea pigs were treated with TRPV4 agonist/antagonist interventions, it was found that the ATP content in the airways of the GSK1016790A group was significantly higher than that of the control group (1.08 ± 0.29 vs. 0.38 ± 0.06, *p* = 0.016). Compared to the CC model, treatment with HC067047 significantly reduced ATP levels in the airways of the CC guinea pigs (0.38 ± 0.06 vs. 1.36 ± 0.12, *p* = 0.009) (Figure 1E). We concluded that the TRPV4 receptor had a certain impact on ATP release in the airways of the guinea pigs. In addition, we found that the ATP levels of the A317491-treated model and probenecid-treated model groups were decreased when compared to the model group (0.52 ± 0.19 vs. 1.36 ± 0.12, *p* = 0.001; 0.51 ± 0.11 vs. 1.36 ± 0.12, *p* = 0.001).

### 3.3. Morphological and Inflammatory Changes in Tracheal Mucosa in Different Drug Groups

By HE staining, we observed that the tracheal mucosa in the control group and the saline-treated control group was smooth and intact, with no significant inflammatory cell infiltration observed. However, the CC group and GSK1016790A-treated control group exhibited thickened tracheal mucosa, accompanied by inflammatory cell infiltration (Figure 1F). Further inflammation evaluation similarly confirmed that the tracheal mucosa inflammation scores were significantly increased in the CC group (4 (3–4) vs. 0 (0–0.5), *p* < 0.001) and the GSK1016790A-treated control group (5 (3.5–5) vs. 0 (0–0.5), *p* < 0.001), compared to the control group. Compared with CC group, the tracheal mucosa inflammation scores were significantly decreased in the HC067047-treated model group (2 (1.5–2) vs. 4 (3–4), *p* = 0.001), the A317491-treated model group (1 (1–2) vs. 4 (3–4), *p* < 0.001), the PSB12062-treated model group (1 (1–2.5) vs. 4 (3–4), *p* < 0.001), and the A804598-treated model group (2 (1–2) vs. 4 (3–4), *p* < 0.001) (Figure 1G).

The results of ELISA showed that compared with the control group, the levels of SP and CGRP in the BALF were significantly increased in both the GSK1016790A-treated control group (SP: 223.96 (209.89–264.38) vs. 175.43 (167.11–184.19), *p* = 0.030; CGRP: 114.43 (92.29–138.28) vs. 79.10 (71.33–86.73), *p* = 0.034) and the CC group (SP: 216.68 (206.14–244.04) vs. 175.43 (167.11–184.19), *p* = 0.013; CGRP:109.51 (103.27–121.55) vs. 79.10 (71.33–86.73), *p* = 0.002). Compared with the CC group, the levels of SP and CGRP in the BALF were reduced to some degree in the HC067047-treated model group, the A317491-treated model group, the PSB12062-treated model group, and the A804598-treated model group (Figure 1H).

Based on the results, we concluded that TRPV4 agonist and citric acid inhalation increased the levels of SP and CGRP in the BALF of the control guinea pigs, thereby enhancing airway neurogenic inflammation. In contrast, administration of TRPV4 antagonist, as well as P2X3, P2X4, and P2X7 antagonists, could reduce the levels of SP and CGRP in the BALF of the guinea pigs with CC, thereby alleviating airway neurogenic inflammation.

### 3.4. Co-Expression of TRPV4 and P2X3/4/7 Receptors in the Tracheal Carina and Vagal Nerve Tissues

Immunofluorescence multicolor staining results revealed that in the tracheal mucosa, TRPV4 was expressed in epithelial cells and nerve fibers, while P2X3, P2X4, and P2X7 receptors were exclusively expressed in nerve fibers. In the vagal ganglia, TRPV4 was widely expressed in neurons and glial cells, whereas P2X3, P2X4, and P2X7 receptors were expressed only in neurons. The expression of TRPV4 in nerve fibers of the tracheal mucosa (labeled by PG9.5) or vagal neurons (labeled by NeuN) was found to be in close proximity to the expression sites of P2X3, P2X4, and P2X7 receptors (Figure 2A,B).

### 3.5. Expression and Distribution of TRPV4, P2X3, P2X4, and P2X7 in the Tracheal Carina and Vagal Nerve Tissues Across Different Drug Groups

#### 3.5.1. Up-Regulation of P2X3, P2X4, and P2X7 Receptor Expression Mediated by TRPV4 in Tracheal Carina and Vagal Ganglia Tissues

Figure 3A and Figure 4A show that in tracheal carina and vagal nerve tissues, the expression of P2X3, P2X4, and P2X7 proteins in both the GSK1016790A-treated control group and the CC group was significantly higher than in the control group. These results suggest that the airway hyper-responsiveness in the CC model guinea pigs is closely related to the increased expression of TRPV4, P2X3, P2X4, and P2X7 proteins, and that TRPV4 may induce the up-regulation of P2X3/4/7 expression.

Further IHC and semi-quantitative analysis revealed that, compared to the control group, the expression levels of P2X3, P2X4, and P2X7 were significantly up-regulated in the tracheal carina and vagal ganglia tissues of the CC group. This result is consistent with Western blot findings. After pretreatment with nebulized GSK1016790A, we observed a significant increase in the distribution and expression of P2X3, P2X4, and P2X7 in the tracheal carina and vagal ganglia tissues of the control group guinea pigs. This change was similar to that observed in the CC group, and no significant differences in the expression of P2X3, P2X4, and P2X7 were found between the two groups. In the CC group, after treatment with HC067047, we observed a significant decrease in the expression of P2X3, P2X4, and P2X7 in the tracheal carina and vagal ganglia tissues. There was no significant difference compared to the control group. These results confirm that TRPV4 mediates changes in P2X3, P2X4, and P2X7 expression. When both TRPV4 antagonist and BzATP were pretreated in the CC group, we found that, compared to the HC067047-only group, the expression of P2X3, P2X4, and P2X7 in the tracheal carina and vagal ganglia was increased in the HC067047+BzATP group. This suggests that BzATP can reverse the TRPV4-mediated down-regulation of P2X3, P2X4, and P2X7.

#### 3.5.2. Down-Regulation of P2X Receptor Expression in TRPV4 Knockout Mice

After performing TRPV4 gene knockout in mice and a 14-day citric acid inhalation pre-treatment, we found that, compared to wild-type mice with citric acid inhalation, the expression and distribution of P2X3, P2X4, and P2X7 proteins in the tracheal carina mucosa of TRPV4 knockout mice were significantly reduced (Figure 5A). This suggests that the TRPV4-P2X pathway plays an important role in mediating the citric acid-induced cough mechanism.

#### 3.5.3. TRPV4 Mediates the Expression of PANX1 in the Tracheal Carina and Vagal Ganglia Tissues

IHC and semi-quantitative analysis revealed that PANX1 expression in the carina and ganglia tissues of the HC067047-treated model group was significantly reduced compared to the model group. However, when the model group was subjected to combined intervention with the HC067047 and BzATP, PANX1 expression in the carina and ganglia tissues showed no significant change compared to the HC067047-treated model group (Figure 5B). Combined with the previous finding that TRPV4 affects ATP levels, this suggests that a TRPV4-mediated increase in PANX1 expression in the tracheal carina tissue and vagal ganglia may be a mechanism for ATP release.

### 3.6. Electrophysiological Validation of the TRPV4-Mediated Purinergic Signaling Pathway

#### 3.6.1. Primary Culture and Identification of Guinea Pig Nodose Ganglion Cells

The morphological changes in the vagal neurons during the culture process are shown in Figure 6A. Immunofluorescence staining using the neuronal marker antibody NeuN and the nuclear dye DAPI revealed that the cells marked with green fluorescence were nodose ganglion (the important part of the vagal ganglia) neurons (Figure 6B).

#### 3.6.2. TRPV4-Mediated Calcium Influx

We performed calcium imaging on the neurons from both the control group and the CC group. The results showed that, in the control group and the CC group, GSK1016790A significantly reduced Ca^2+^ concentration in the nodose ganglion neurons (*p* = 0.043; *p* = 0.013). HC067047 induced an increase in the Ca^2+^ concentration in the nodose ganglion neurons (*p* = 0.002; *p* = 0.004). Furthermore, the Ca^2+^ concentration in the CC+GSK1016790A group was significantly increased compared to that in the control+GSK1016790A group (*p* = 0.044) (Figure 6C). These results indicate that the Ca^2+^ concentration in the vagal neurons is affected by TRPV4 receptors, and neurons from the CC group exhibited higher excitability compared to the control group and were more sensitive to changes in TRPV4 channel activity.

#### 3.6.3. Whole-Cell Patch Clamp

To further verify the relationship between TRPV4 and ATP-P2X, we performed whole-cell patch clamp experiments on the nodose ganglion neurons from both the control and CC groups. The results showed that, when ATP (10^−4^ M) was applied extracellularly to neurons in both groups, a significant inward current (I_ATP_/pA) was recorded. Compared to the control group, I_ATP_ was significantly higher in the CC group (451.00 ± 33.72 vs. 297.75 ± 32.15, *p* = 0.001), with an amplitude of 153.24 ± 18.11% of the I_ATP_ in the control group (*n* = 5) (Appendix A). In the control group, when A317491 (10 μM: 124.25 ± 16.45 vs. 297.75 ± 32.15, *p* < 0.001), PSB12062 (30 μM: 162.25 ± 5.56 vs. 297.75 ± 32.15, *p* < 0.001), and A804598 (8 μM: 146.00 ± 2.94 vs. 297.75 ± 32.15, *p* < 0.001) were applied, the I_ATP_ amplitude was significantly reduced (Figure 6D). Similarly, in the CC group, the I_ATP_ inward current was also significantly reduced by A317491 (10 uM: 277.50 ± 26.38 vs. 451.00 ± 33.72, *p* < 0.001), PSB12062 (30 μM: 263.75 ± 13.67 vs. 451.00 ± 33.72, *p* < 0.001), and A804598 (8 μM: 236.75 ± 20.54 vs. 451.00 ± 33.72, *p* < 0.001) (Figure 6D). The I_ATP_ amplitudes of the control and CC groups were shown in Appendix A. The above results suggest that the inward current induced by ATP was higher in the CC group and can be blocked to varying degrees by P2X3, P2X4, and P2X7 antagonists.

When the nodose ganglion neurons of the CC group were pretreated with GSK1016790A (100 nM) or HC067047 (1 μM) for 24 h and washed out, I_ATP_ measurements were conducted. The results showed that the I_ATP_ in the CC+GSK1016790A group was significantly higher compared to the CC group (626.00 ± 52.37 vs. 399.50 ± 60.67, *p* = 0.004) (Figure 6E), with the enhanced I_ATP_ amplitude being 154.73 ± 29.39% of the CC group (*n* = 5) (Appendix A). In contrast, the CC+HC067047 group showed a significant reduction in I_ATP_ compared to the CC group (230.50 ± 19.80 vs. 591.25 ± 20.15, *p* < 0.001) (Figure 6E), with a decrease in amplitude of 61.86 ± 7.15% (*n* = 5) (Appendix A). Furthermore, the inhibition of I_ATP_ by HC067047 was irreversible. Herein, we concluded that TRPV4 may modulate ATP-activated currents.

When the nodose ganglion neurons of the CC group were incubated with GSK1016790A (100 nM) for 24 h, followed by treatment with A317491 (10 μM), PSB12062 (30 uM), or A804598 (8 μM), we observed that, compared to the GSK1016790A group, the I_ATP_ levels in GSK1016790A+A317491 group (153.25 ± 8.65 vs. 591.25 ± 0.15, *p* < 0.001), GSK1016790A+PSB12062 group (212.75 ± 27.80 vs. 591.25 ± 20.15, *p* < 0.001) (Figure 6E), and GSK1016790A+A804598 group (210.50 ± 9.98 vs. 591.25 ± 20.15, *p* < 0.001) were significantly reduced. The reduced I_ATP_ amplitudes were 26.26 ± 2.07%, 36.93 ± 5.91%, and 36.46 ± 2.62% of that of the TRPV4 agonist group, respectively (*n* = 5) (Appendix A). The results indicate that TRPV4 agonists can enhance ATP-activated currents, and the enhanced currents can still be inhibited by P2X3, P2X4, and P2X7 receptor antagonists; furthermore, when the combination of P2X3, P2X4, and P2X7 receptor antagonists was used as a treatment in the TRPV4 agonist group, the inhibited efficiency was maximal. The I_ATP_ amplitude in the CC+GSK1016790A+A317491/PSB12062/A804598 group was 13.16 ± 2.13%, compared to 19.96 ± 0.70% in the CC group (Appendix A).

## 4. Discussion

The pathophysiological basis of cough reflex hypersensitivity lies in neuro-inflammation and hyper-excitability of the vagal nerve pathway, which may occur in the central nervous system, the peripheral nervous system, or both simultaneously [34,35]. Peripheral sensitization interacts with and influences central sensitization [36,37]. The primary mechanism of peripheral sensitization is the activation of sensory nerve fibers, specifically the activation or increased sensitivity of Aδ fibers and C fibers, leading to enhanced sensory input and increased peripheral sensitization [38]. The TRP family, expressed in primary sensory nerve fibers and neurons of the respiratory tract, and functioning as molecular integrators of noxious stimuli, is believed to be closely associated with cough reflex hypersensitivity [39,40,41].

Currently, there is limited research and exploration regarding the role of TRPV4 in the treatment of RCC. Pre-clinical studies have provided evidence that TRPV4 agonist GSK1016790A can induce coughing in healthy guinea pigs, and is inhibited by the selective TRPV4 antagonist HC067047. The mechanism may be related to vagal nerve depolarization [18]. Buday et al. found that TRPV4 antagonists partially inhibited citric acid-induced coughing in guinea pigs, which is related to TRPV4-mediated activation of airway afferent nerve fibers [42]. However, in a recent clinical trial, Ludbrook et al. indicated that treatment with the TRPV4 antagonist GSK2798745 did not produce any anti-cough effect when compared to a placebo [43]. This could be due to the fact that the dose was not high enough for adequate receptor occupancy [10], and the conclusion was made based on 16 participants. Overall, there are few clinical studies on the use of TRPV4 antagonists for the treatment of coughs, and the number of cases included in these studies, as well as the types of antagonists investigated, is limited. Based on the mechanisms and the existing controversy regarding its clinical outcomes, it is crucial to clarify the role and mechanism of the TRPV4 receptor in cough hypersensitivity.

Based on our previous study [31], we have established and validated a citric acid-induced chronic cough guinea pig model. In this model, inhalation of citric acid not only enhanced cough hypersensitivity, but also increased the proportion of neutrophils in the airways and the release of mediators involved in neurogenic inflammation, such as SP and CGRP. This was further associated with damage to the airway epithelium, thickening of the basement membrane, ciliary dysfunction and shedding, infiltration of inflammatory cells, and an increase in mucus within the airway lumen. These findings suggest that a model of airway inflammation-related chronic cough which closely mirrors the pathological features of chronic cough in humans was successfully established.

Further studies based on this model have observed the effects of the TRPV4 receptor on cough hypersensitivity in guinea pigs with CC. The results found that TRPV4 agonists significantly increased cough reactivity in the control group, and the TRPV4 antagonist significantly reduced cough reactivity in the CC group, which suggests that the TRPV4 receptor plays a role in both acute and chronic cough in guinea pigs, and contributes to the development of cough hypersensitivity. Further research has indicated that the TRPV4 receptor influences changes in neurogenic inflammatory neuropeptides (SP and CGRP) and airway mucosa. Neurogenic inflammation is an important mechanism underlying cough hypersensitivity [44,45]. When primary sensory nerve fibers in the respiratory tract and lungs are stimulated and activated, neuropeptides such as SP and CGRP are released from afferent nerves, inducing neurogenic inflammatory reactivity [46]. On the one hand, neuropeptides can directly act on irritant receptors to trigger the cough reflex. On the other hand, they can induce neurogenic inflammation, altering the phenotype and excitability of airway sensory afferent nerves. This process leads to plasticity in both peripheral and central neurophysiology [47]. These findings suggest that TRPV4 may mediate the generation and persistence of inflammation in the development of cough hypersensitivity. Additionally, our study also found that the TRPV4 receptor has a certain impact on ATP release in the airways of guinea pigs. ATP, as a short-term signaling molecule, has been shown to play a role in various diseases, including visceral pain, bladder incontinence, hypertension, and CC [48,49,50,51]. ATP can be released from non-neuronal cells, such as damaged airway epithelium and immune cells, via cell lysis or PANX1 ion channels. In a chronic pain model, PANX1-dependent ATP release in the dorsal root ganglion and trigeminal nerve was increased, contributing to the development of abnormal pain perception [52,53]. Our study also confirmed that TRPV4 can influence changes in ATP concentration within the airways, along with alterations in PANX1 expression. Further study of the PANX1 antagonist (probenecid) demonstrated that probenecid significantly reduced ATP levels in the airways of CC guinea pigs, confirming that ATP drives cough hypersensitivity. Therefore, we hypothesize that in the guinea pigs of the citric acid-induced enhanced cough model, TRPV4-mediated ATP release may have depended on the activation of PANX1 channels, contributing to the development of cough sensitivity. The potential mechanism involves TRPV4 agonists inducing a concentration-dependent increase in Ca^2+^ levels in the vagal ganglia, with Ca^2+^ influx activating ATP release through PANX1 ion channels [54]. In the future, well-designed experiments are still needed to further verify whether the knockout or over-expression of PANX1 channels in a citric acid-induced enhanced cough model reduces heightened cough sensitivity, affects ATP release, and alters calcium level changes. Released ATP binding to P2X might stimulate afferent sensory nerves and promote the release of other pro-inflammatory cytokines, driving further neurogenic inflammation [55,56] and contributing to the chronic sensitization of peripheral nerves. Thus, these findings suggest that TRPV4 may be involved in the development of cough hypersensitivity, potentially in relation to changes in ATP levels.

TRPV4 is a non-selective cation channel with high permeability to Ca^2+^, and changes in intracellular and extracellular Ca^2+^ levels can regulate TRPV4 function [57,58]. Ca^2+^ channels play various roles in the nervous system, such as regulating neuronal excitability, neurotransmitter release at synaptic sites, synaptic plasticity, and gene transcription [59]. In pathological neuropathic pain models, TRPV4 channels on dorsal root ganglion cells can be modulated by changes in intracellular and extracellular Ca^2+^ concentrations, playing an important role in nociceptive sensitization [60]. Additionally, TRPV4 activation can promote Ca^2+^ influx and is involved in the molecular signaling pathway of TRPV4-NO-mediated nociceptive hypersensitivity in rat neuropathic pain models [61]. In this study, we also analyzed the Ca^2+^ concentration in the nodose ganglion cells, and found that the Ca^2+^ concentration in the nodose ganglion neurons of both the healthy and CC groups could be influenced by TRPV4; specifically, there were TRPV4-dependent changes in Ca^2+^ concentration. Moreover, the Ca^2+^ concentration in the model group was higher, suggesting that the nodose ganglion neurons in the CC guinea pig model had higher basal excitability.

Purinergic P2X receptors are ligand-gated, non-selective cation channels that are selectively sensitive to ATP, and play a regulatory role in peripheral nerve sensitization. P2X receptors are currently believed to be associated with neuropathic pain, pruritus, and CC [39,50]. We systematically studied the effects of P2X1-7 receptor antagonists on cough hypersensitivity, and found that nebulized inhalation of P2X3, P2X4, and P2X7 receptor antagonists reduced cough reactivity in the CC group to varying degrees, which is consistent with previous studies on the roles of P2X4 and P2X7 in regulating chronic pain [25,26,62,63]. However, there was no significant difference in guinea pigs of the model treated with P2X1 or P2X2 receptor antagonists compared to the CC model group. This indicates that P2X3, P2X4, and P2X7 receptors play a role in chronic cough hypersensitivity. Further studies revealed that P2X3, P2X4, and P2X7 receptor antagonists can influence neurogenic inflammation and airway mucosal integrity, which needs to be explored in future study.

To explore the mechanism of activation of ATP-P2X receptor channels by TRPV4 in cough hypersensitivity, we further performed related experiments at the molecular and electrophysiological levels. Western blotting and immunohistochemical analysis revealed that TRPV4 mediated the expression and distribution of P2X3/4/7, involving ATP regulation, with the most noticeable changes occurring in the vagal ganglia. Under hypotonic conditions, it has been validated that TRPV4 mediates P2X3 receptor activation, inducing airway sensory nerve depolarization and cough sensitization [18]. Our results similarly confirm at the protein level that TRPV4 mediates changes in P2X3 protein expression, suggesting that TRPV4 can alter the number and sensitivity of P2X3 receptors. In chronic neuropathic pain models, Fang et al. [64] found that after blocking TRPV4 with the antagonists RN-1734 and HC-067047, P2X7 expression in the dorsal root ganglion decreased, demonstrating that TRPV4-mediated P2X7 expression participates in the development of neuropathic pain. This conclusion is consistent with our findings regarding the CC mechanism. TRPV4 activation increases intracellular Ca^2+^ and releases ATP from ASM cells, triggering the release of P2X4-dependent cysteinyl leukotrienes from mast cells, leading to airway constriction [65,66]. In our study, we similarly confirmed, in the CC model, TRPV4-mediated P2X4 expression in the airway nerve fibers and vagal ganglia tissues.

The vagal ganglion serves as the primary sensory neuron for sensing and transmitting cough impulses, playing a role in cough signal transduction and cough sensitization [67]. ATP, released from damaged airway epithelium and immune cells, activates P2X receptors on sensory neurons, leading to the transmission of cough-related sensory information from the periphery to the central nervous system, ultimately triggering the cough reflex [49,68,69]. Studies have shown that TRPV4 channels have modulatory effects on various ion channel currents, including TTX-sensitive and TTX-resistant sodium channels, as well as potassium and Ca^2+^ channels [70]. However, whether TRPV4 channels modulate ATP-activated currents remains unclear. By whole-cell patch clamp techniques, our research aimed to investigate whether TRPV4 channels can modulate ATP-activated currents in vagal ganglion cells of guinea pigs, thereby providing electrophysiological evidence for the involvement of the TRPV4-mediated purinergic signaling pathway in the development of cough hypersensitivity in CC guinea pigs. When extracellular ATP was applied to neurons, most of the tested cells of the control and CC groups exhibited a significant inward current. Notably, the ATP-induced inward current was higher in the CC group compared to the control group. This inward current could be partially blocked by P2X3, P2X4, and P2X7 receptor antagonists, suggesting that exogenous ATP activates P2X3/4/7 receptor-mediated currents in nodose ganglion cells. This finding aligns with previous studies involving chronic pain models [71,72,73]. The enhanced inward current in the nodose ganglion cells of the CC guinea pigs further corroborates the increased number and sensitivity of P2X3, P2X4, and P2X7 receptors. To examine the modulatory role of TRPV4 on ATP-activated currents, nodose ganglion neurons from the CC group were pre-incubated with TRPV4 agonists or antagonists for 24 h before ATP testing (I_ATP_) was conducted. The results showed that TRPV4 agonists enhanced ATP-activated currents, and this enhancement could still be inhibited by P2X3, P2X4, and P2X7 receptor antagonists, with the combined inhibition of P2X3/4/7 being the most effective. Conversely, TRPV4 antagonists partially suppressed ATP-activated currents, and this inhibitory effect was irreversible. These findings suggest that TRPV4 modulates ATP-activated currents mediated by P2X3/4/7 receptors, and plays a critical role in the development of cough hypersensitivity. The underlying mechanism may involve TRPV4 directly or indirectly regulating P2X receptor expression or function through ATP, thereby influencing ATP-activated currents. This modulation alters the sensitivity and excitability of the nodose ganglion to cough stimuli, enhancing or suppressing the ascending transmission of cough signals.

## 5. Conclusions

In summary, this study explored the mechanism of the TRPV4-mediated purinergic signaling pathway in cough hypersensitivity from the perspective of peripheral sensitization in vivo and in vitro. These findings provide a potential research direction for the development of therapeutics targeting RCC.

## Figures and Tables

**Figure 1 biomolecules-15-00285-f001:**
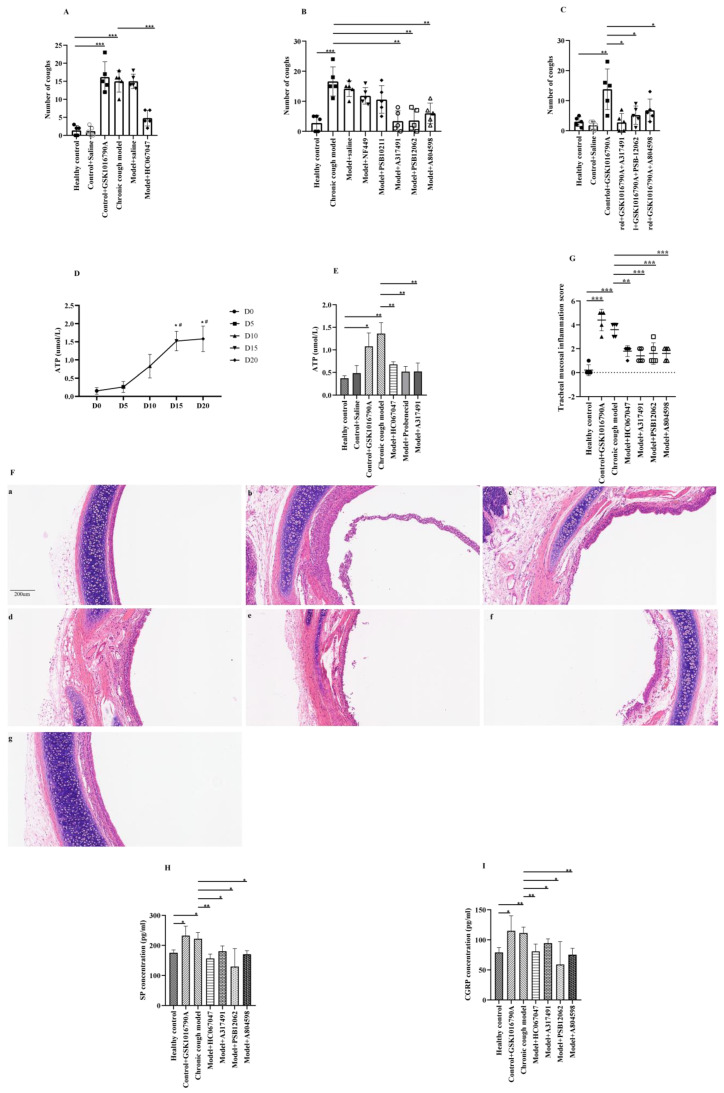
The effects of pharmacological modulation of TRPV4 and P2X receptors on cough reactivity, ATP level, and morphological and inflammatory changes. (**A**–**C**): The effects of different drug interventions on cough reactivity in guinea pigs with chronic cough; (**D**,**E**): changes in ATP in the airway of guinea pigs; (**F**,**G**): morphological changes in the tracheal carina in the chronic cough model guinea pigs under TRPV4 and P2X3,4,7 inhibitors (×100). (**a**) Healthy control group; (**b**) control+GSK1016790A group; (**c**) chronic cough model group; (**d**) model+HC067047 group; (**e**) model+A317491 group; (**f**) model+PSB12062 group; (**g**) model+A804598 group. (**H**,**I**): CGRP and substance P in the bronchoalveolar lavage fluid of guinea pigs under different drug interventions * *p* < 0.05, ** *p* < 0.01, *** *p* < 0.001.

**Figure 2 biomolecules-15-00285-f002:**
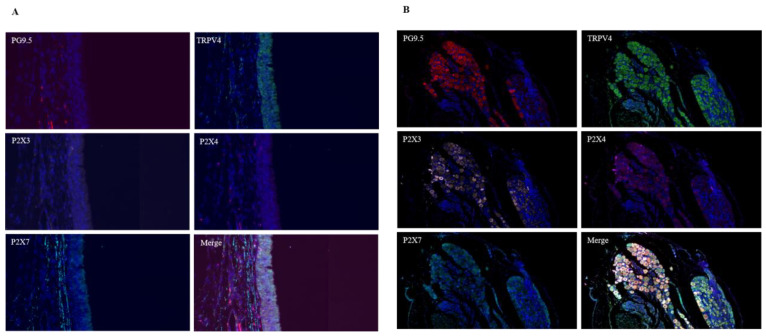
Co-expression of the PGP9.5, TRPV4 receptor, and P2X3,4,7 receptors in the tracheal carina and vagal ganglion tissues. (**A**): Tracheal carina; (**B**): vagal ganglion. Red represents the PGP9.5 receptor; green represents the TRPV4 receptor; pink represents the P2X3 receptor; magenta represents the P2X4 receptor; and blue represents the P2X7 receptor. Merged image, ×400.

**Figure 3 biomolecules-15-00285-f003:**
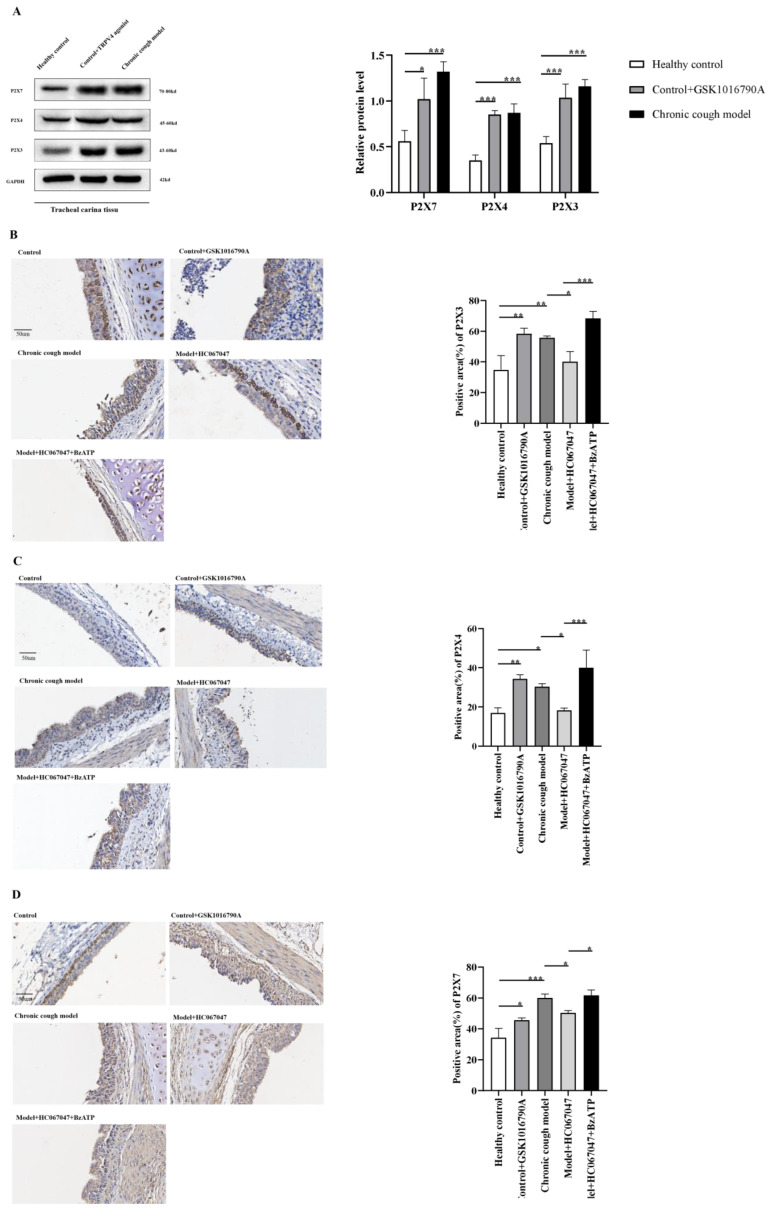
TRPV4-mediated up-regulation of P2X3, P2X4, and P2X7 receptors, mediated by TRPV4, in the tracheal carina and vagal ganglia tissues. (**A**): The expression of TRPV4, P2X3, P2X4, and P2X7 proteins in the tracheal carina in different drug groups. (**B**–**D**): The distribution of P2X3, P2X4, and P2X7 in the tracheal carina in different drug groups, respectively. Merged image, ×400. * *p* < 0.05, ** *p* < 0.01, *** *p* < 0.001.

**Figure 4 biomolecules-15-00285-f004:**
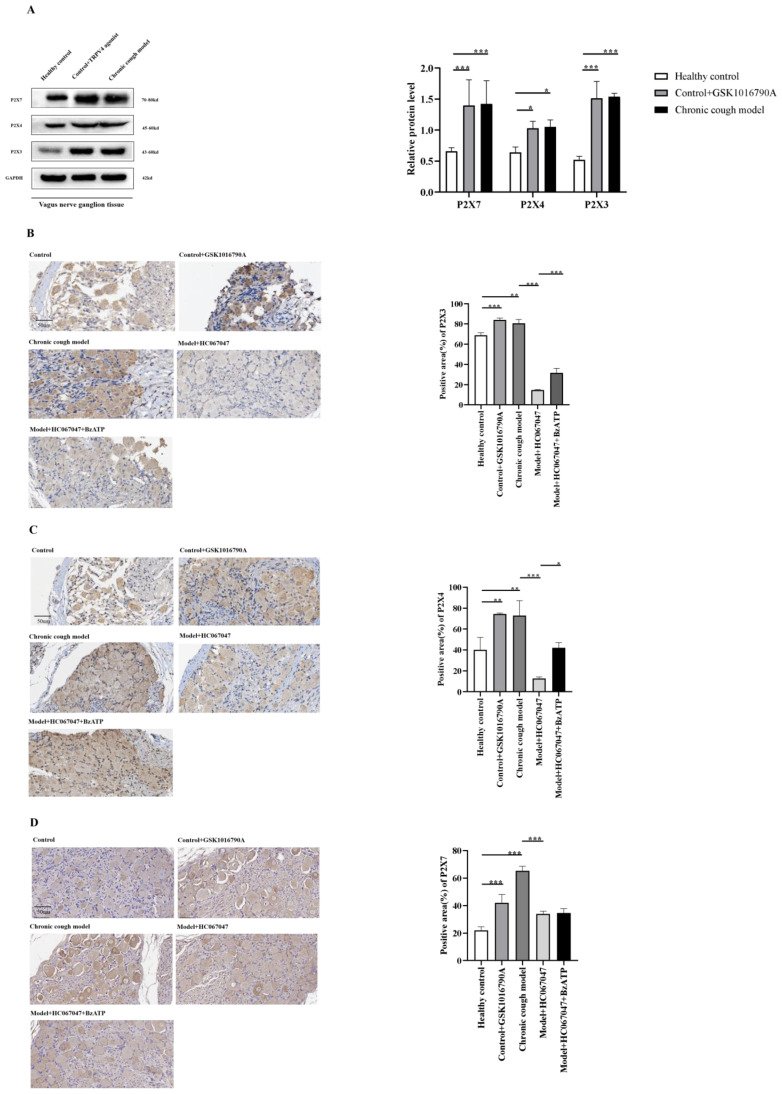
TRPV4-mediated up-regulation of P2X3, P2X4, and P2X7 receptors in vagal ganglia tissues. (**A**): The expression of TRPV4, P2X3, P2X4, and P2X7 proteins in the vagal ganglia in the different drug groups. (**B**–**D**): The distribution of P2X3, P2X4, and P2X7 in the vagal ganglia in the different drug groups, respectively. Merged image, ×400. * *p* < 0.05, ** *p* < 0.01, *** *p* < 0.001.

**Figure 5 biomolecules-15-00285-f005:**
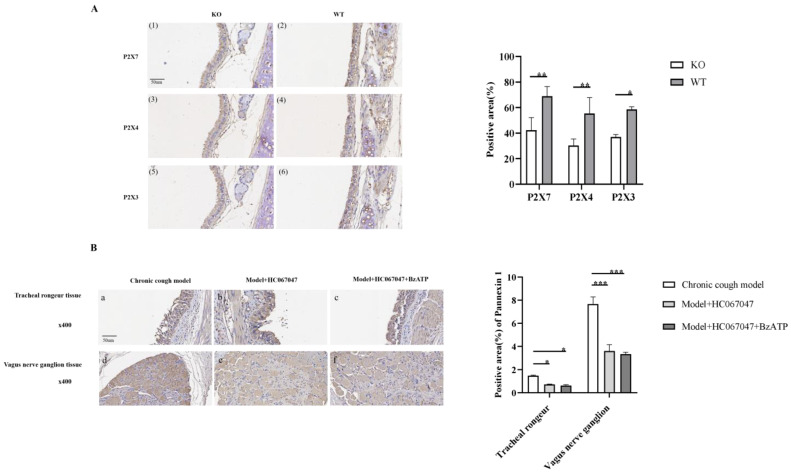
Immunohistochemistry comparison under different interventions. (**A**): Down-regulation of P2X receptor expression in TRPV4 knockout mice, merged image, ×400, (**1**,**2**) represent P2X7 expression, (**3**,**4**) represent P2X4 expression, (**5**,**6**) is P2X3 expression; (**B**): immunohistochemistry comparison of Pannexin 1 expression, merged image, ×400, (**a**–**c**) are immunohistochemical staining of tracheal carina tissue; (**d**–**f**) are immunohistochemical staining of vagal ganglion tissue. * *p* < 0.05, ** *p* < 0.01, *** *p* < 0.001.

**Figure 6 biomolecules-15-00285-f006:**
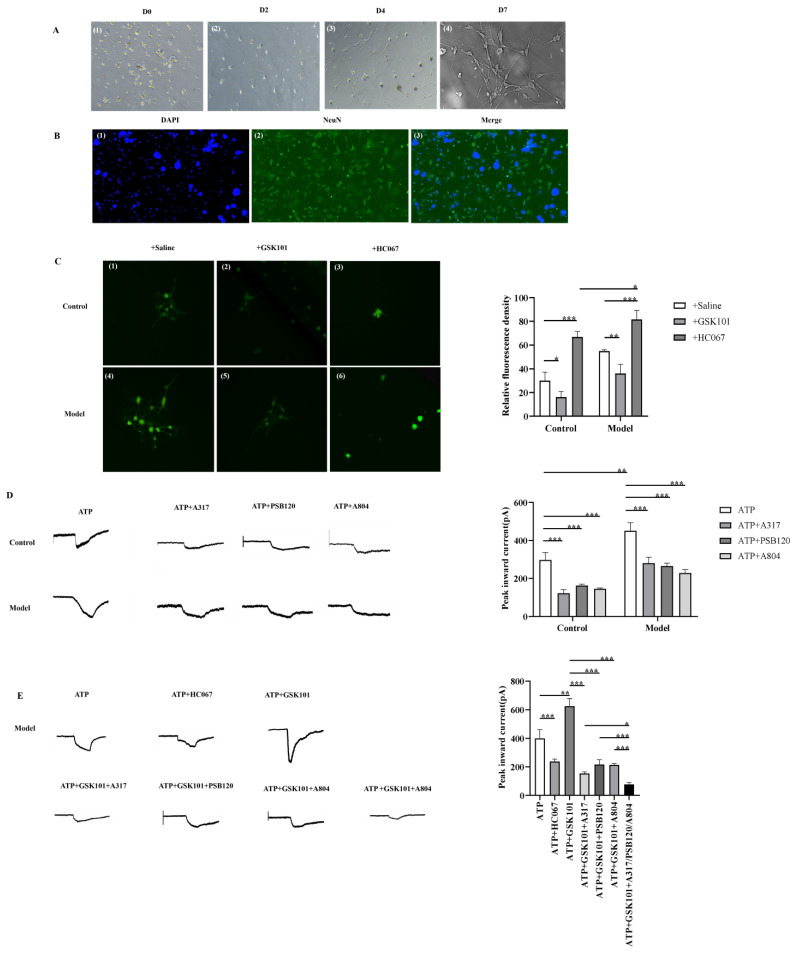
Electrophysiological evidence of TRPV4-mediated purinergic signaling pathway. (**A**,**B**): Primary culture and identification of guinea pig nodose ganglion cells, D0–D7 refers to observations under a light microscope from day 0 to day 7, (**1**) DAPI-stained nuclear image, (**2**) NEUN-stained image, (**3**) Merged image. (**C**): Calcium imaging: TRPV4-mediated calcium influx. (**D**,**E**): Pharmacological modulation of whole-cell ATP-induced currents. * *p* < 0.05; ** *p* < 0.01; *** *p* < 0.001.

## Data Availability

The datasets generated for this study are available upon request from the corresponding author.

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
