# Peer review of "Activation of Adenosine Triphosphate-Gated Purinergic 2 Receptor Channels by Transient Receptor Potential Vanilloid Subtype 4 in Cough Hypersensitivity"

_biomolecules, 2025, doi:10.3390/biom15020285_

Round 1
Reviewer 1 Report
Comments and Suggestions for Authors
In this study, Li et al. investigate the role of TRPV4 and ATP P2X receptors in cough hypersensitivity using a guinea pig model. They performed pharmacological experiments in vitro and in vivo, combined with immunochemical and electrophysiological approaches in a model of citric acid-induced chronic cough. They found that TRPV4 and P2X3, P2X4, P2X7 receptors contribute to cough hypersensitivity. TRPV4 activation increases cough and ATP release, while antagonists reduce these effects. The study concludes that the TRPV4-ATP-P2X3/4/7 pathway is critical for the development of cough hypersensitivity. Overall, the data support the conclusions, but major concerns should be addressed before publication.
Major concerns:
- In Figure 6E, the ATP+GSK101 current recording looks very leaky, as it does not show the corresponding kinetics for the development of the ATP-induced current, but something very sharp. In addition, there is no recovery of the basal current, suggesting a low-quality recording that may overestimate the measured current. Based on this recording, I do not think it can be concluded that GSK101 potentiates the ATP-induced current in cultured vagal neurons.
- Line 229: It is very important to include a data analysis section with a description of the statistics used.
Minor concerns:
- Title: I would recommend not using TRPV4-ATP-P2X together because it looks like the name of a single ion channel rather than a pathway involving two different channels. Authors should change this in the title.
- Line 19: there is a change in font size
- Lines 25 to 29: There are data that are not necessary in the abstract because they are in the results section. Authors should remove them from the abstract to avoid repetition.
- Line 90: Neither Ca2+ nor CGRP are cytokines. Authors should correct this wherever it appears.
- Line 114: "to explore the mechanism of TRPV4 activity" doesn't make sense in the context of this paper, since the authors are not investigating such mechanisms, but the involvement of TRPV4 in the ATP-P2X pathway.
- Line 121: I don't understand the meaning of "recommended" in this context. Also "penta orbital"?
- Line 140: I would say "these procedures" instead of "this experiment".
- Line 240: "P2X74"?
- Line 337: "Ca2+ detection" is incorrect. Should be "calcium imaging”
- Line 347: delete "detection
- Line 352: add current units to the text
- Lines 407-408: SP and CGRP are not cytokines, they are neuropeptides!
- Line 455: What is WB and IHC?
- Line 679: "The effects of TRPV4 and P2X receptors on cough reactivity" is incorrect. I recommend "The effects of pharmacological modulation of TRPV4 and P2X receptors on cough reactivity".
- Line 693: "Expression upregulation"? That doesn't make sense. It should be something like "TRPV4-mediated up-regulation of P2X3,..."
- Line 698: Same as line 693
- Line 704: "Electrophysiological evidence" is not correct. it could be something like "Electrophysiological evidence of .... in vagal neurons".
- Line 707: "Whole cell patch clamp detection: IATP in different drug groups". This is not an adequate description. It might be better to say something like "Pharmacological modulation of whole-cell ATP-induced currents".
Comments on the Quality of English LanguageThis paper needs to improve the quality of English
Reviewer 2 Report
Comments and Suggestions for Authors
Li et al describe a paper with a number of methodologies to demonstrate the role of TRPV4 and ATP receptors in chronic cough. They have utilised a cough model, IHC staining and neuron calcium imaging and patch clamp to determine the role of TRPV4 and ATP in chronic cough. Although elegant and in parts well described, I would suggest some changes prior to publication.
· Unfortunately I have an issue with the chronic cough model mentioned within the manuscript. There is no background or literature to suggest that exposing guinea pigs twice a day to the tussive agent citric acid will induce a ‘chronic cough’ phenotype. We currently do not know what is driving chronic cough, and the authors have provided no reason as to why this method recapitulates a chronic cough phenotype. The authors have not shown the number of coughs to CA over the 15 day period to demonstrate that there is increased responsiveness so therefore I do not feel that the conclusion can be made that it the response is hyperresponsive. Citric acid causes cough on its own and therefore could explain the larger response. The increase in levels of ATP in the BAL could have been through repetitive damage of the epithelium and not through TRPV4 – this would have needed some further pharmacology to prove.
· There is no description of the acute cough model in the methodology. How long was GSK101 administered? How were coughs counting? Was the control group also exposed to CA at that timepoint?
· The figures are far too small and hard to read. These would need enlarging prior to any publication
· There are no controlled experiments for the Western blots to suggest that the antibodies were selective for the receptors they were labelling. Either literature or some workup would be required in the supplementary information to demonstrate selectivity.
· There is no mention of the TRPV4 clinical trial in chronic cough which demonstrated no efficacy on objective cough frequency with an oral TRPV4 inhibitor in patients with chronic cough.
· Methodology is lacking from the neuron calcium imaging experiments. Guinea pigs have a jugular and nodose ganglion – which cells were imaged in the experiments? These provide different phenotypes of nerve fibres and therefore important to distinguish in this particular pathway. Although the patch clamp data is novel and exciting the same issue arises – what population are the cells from?
Round 2
Reviewer 1 Report
Comments and Suggestions for Authors
The authors addressed my concerns in a satisfactory manner.
Author Response
Thank you for your valuable comments and suggestions. We sincerely appreciate the time and effort you have invested in reviewing our manuscript. Your feedback has been instrumental in improving the quality of our work.
Reviewer 2 Report
Comments and Suggestions for Authors
1. I thank the authors to their response to point 1 – and agree that in some cases that the authors have outlined cough is associated with excess airway inflammation, however this is not the case in all cases including that of idiopathic chronic cough. I would therefore suggest that the authors rename the chronic cough model to something such as ‘inflammation induced enhanced cough’ or ‘citric acid induced enhanced cough’ which would better describe the phenotype and stop any misunderstanding in the field.
2. Thank you for including this detail
3. Thank you for the clarification and the inclusion of a control group. It has been documented that a P2X2/3 antagonist could reduce this, and this would add greatly to the authors conclusions (as well as a PANX1 antagonist as suggested by the authors). I would strongly suggest that this experiment is included to demonstrate that ATP is driving the cough hypersensitivity in this model
4. Thank you for the inclusion – this is not quitre the methodology used in that paper (GSK101 was aerosolised for 15 minutes and cough counted for that time period and 5 minutes following aerosolization) but you have included the methodology so my point has been answered.
5. Thank you for increasing the figures
6. Thank you for including the controls
7. This point has been added – great.
8. Thank you for the clarification that these are the nodose neurons that experiments were carried out on – I believe it to be an important addition to the paper
My final point would be to gently suggest proof reading the cover letter prior to posting the response – as the cover letter states that this is a response to “Current Diagnosis and Treatment Options for Non-acid GERC and Future Prospects” which initially caused myself some confusion!
Author Response
Thanks for your suggestions. More details would be shown in Response Letter.
